# Global Fisheries Responses to Culture, Policy and COVID-19 from 2017 to 2020

**Bin He** [1,2], **Fengqin Yan** [2,3,4], **Hao Yu** [2,3,4], **Fenzhen Su** [1,2,3,4,*] , **Vincent Lyne** [3,5] , **Yikun Cui** [2], **Lu Kang** [2,3,4] and **Wenzhou Wu** [2,3]

1  Faculty of Geomatics, Lanzhou Jiaotong University, Lanzhou 730070, China; heb@lreis.ac.cn
2  Collaborative Innovation Center of South China Sea Studies, Nanjing 210023, China; yanfq@lreis.ac.cn (F.Y.); yuh@lreis.ac.cn (H.Y.); cuiyikun@smail.nju.edu.cn (Y.C.); kangl@lreis.ac.cn (L.K.); wuwz@lreis.ac.cn (W.W.)
3  State Key Laboratory of Resources and Environmental Information System, Institute of Geographic Sciences and Natural Resources Research, Chinese Academy of Sciences, Beijing 100101, China; vincent.lyne@utas.edu.au
4  College of Resources and Environment, University of Chinese Academy of Sciences, Beijing 100049, China
5  IMAS-Hobart, University of Tasmania, Hobart, TAS 7004, Australia
*  Correspondence: sufz@lreis.ac.cn

**Abstract:** Global Fishing Watch (GFW) provides global open-source data collected via automated monitoring of vessels to help with sustainable management of fisheries. Limited previous global fishing effort analyses, based on Automatic Identification System (AIS) data (2017–2020), suggest economic and environmental factors have less influence on fisheries than cultural and political events, such as holidays and closures, respectively. As such, restrictions from COVID-19 during 2020 provided an unprecedented opportunity to explore added impacts from COVID-19 restrictions on fishing effort. We analyzed global fishing effort and fishing gear changes (2017–2019) for policy and cultural impacts, and then compared impacts of COVID-19 lockdowns across several countries (i.e., China, Spain, the US, and Japan) in 2020. Our findings showed global fishing effort increased from 2017 to 2019 but decreased by 5.2% in 2020. We found policy had a greater impact on monthly global fishing effort than culture, with Chinese longlines decreasing annually. During the lockdown in 2020, trawling activities dropped sharply, particularly in the coastal areas of China and Spain. Although Japan did not implement an official lockdown, its fishing effort in the coastal areas also decreased sharply. In contrast, fishing in the Gulf of Mexico, not subject to lockdown, reduced its scope of fishing activities, but fishing effort was higher. Our study demonstrates, by including the dimensions of policy and culture in fisheries, that large data may materially assist decision-makers to understand factors influencing fisheries' efforts, and encourage further marine interdisciplinary research. We recommend the lack of data for small-scale Southeast Asian fisheries be addressed to enable future studies of fishing drivers and impacts in this region.

**Keywords:** automatic identification system; COVID-19; fishery policy; fishing culture; fishing gear; global fishing watch; spatio-temporal analysis

## 1. Introduction

Managing global fisheries is complicated by the increasing fishing effort of developed and developing countries. However, the lack of monitoring and rational management of global marine fishery resources is confounded by the extensive number of fishing vessels, limited regulatory forces, the lack of data for small-scale fisheries, and the dynamic and limited characteristics describing marine fisheries [1–3]. Recent developments in large data collected from the Automatic Identification System (AIS) [4–6] now provides dynamic vessel information and local ocean environment data, which is used for vessel safety [7–9], environmental pollution [10–12] and traffic monitoring [13–15]. Other data uses include the following: Marine Spatial Planning [16,17] research; Longépé et al. (2018) combined

Vessel Detection System (VDS) and AIS data for monitoring real-time illegal fishing [18]; Ferrà et al. (2018) mapped trawling activity for two consecutive years [19]; Liu et al. (2019) showed high fishing levels throughout the year in the Antarctic Peninsula [20]; and Russo et al. (2020) identified spatio-temporal changes in fishing effort and fleet dynamics [21]. Despite this range of fishing vessels impact studies, limited fisheries research on culture and policy [22] suggest these factors more strongly influence fishing effort than common controls associated with economics and environmental drivers. Thus, the goal of this study is to quantitatively explore the details of how culture, policy and the added impact from COVID-19 lockdown affected global fishing effort.

In 2020, the spread of COVID-19 affected the world's society [23–25], economy [26–28] and environment [29–31]. The global scientific community promptly deployed information technology for continuous monitoring of COVID-19 spread, and applications of GIS using comprehensive data [32,33]. Progressive restrictions of activities also impacted fishery operations and catches [34,35], fishery trade [36] and port activities [37]. Although Pellegrini et al. (2020) derived and analyzed the environmental and socio-economic effects of COVID-19 on marine realms [38], quantitative studies describing impacts on global fisheries and spatio-temporal dynamics are still lacking.

Thus, the objectives of this study were: (1) to determine the spatio-temporal changes of global fishing effort and fishing gear from 2017 to 2020; (2) to quantify the cultural and policy drivers influencing fishing effort and fishing gear through GIS spatial analyses and statistics (2017–2019); and (3) to analyze the effects of lockdown policies on fishing effort of a few representative countries (China, Spain, Japan and the US) in 2020.

## 2. Materials and Methods

### 2.1. Study Area and Data

The study was divided into two parts (Figure 1). Firstly, a global scale study using AIS data from GFW was used to examine the impact of culture and policy on fishing effort from 2017 to 2019. Secondly, to highlight the influence of the COVID-19 lockdown policies on fisheries, we analyzed fishing effort of four typical countries (China, Spain, Japan and the US). China, Spain and Japan were selected because they account for a relatively high proportion of the GFW data, and their fishery activities are typical and representative of the Food and Agriculture Organization (FAO) statistics over the years. The United States was selected because some States implemented lockdown during the period from March−May 2020; the United States was not in national lockdown, and only some states implemented lockdown. For the analyses, we selected the offshore waters of China, Spain and eastern United States, which implemented lockdowns, and Japan, which did not officially implement a lockdown.

Global fisheries data from 2017 to 2020 were retrieved from the website of GFW (https://globalfishingwatch.force.com/gfw/s/datadownload). AIS is composed of static data and dynamic data. The static data includes Maritime Mobile Service Identity, IMO (International Maritime Organization) call sign, ship name, ship type, country of registration, ship length, ship width, draft and purpose. The dynamic data include waypoint time, longitude, latitude, heading over the ground and speed over ground. The GFW AIS data were verified by a machine learning algorithm that combined the AIS dynamic longitude and latitude data with static national data. This provided daily aggregated data of fishing effort and vessel presence per grid cell of $0.01 \times 0.01$ decimal degrees, with effort measured in units of hours. We used the date, gear type and fishing hours fields from the tabular data for the spatio-temporal analysis.

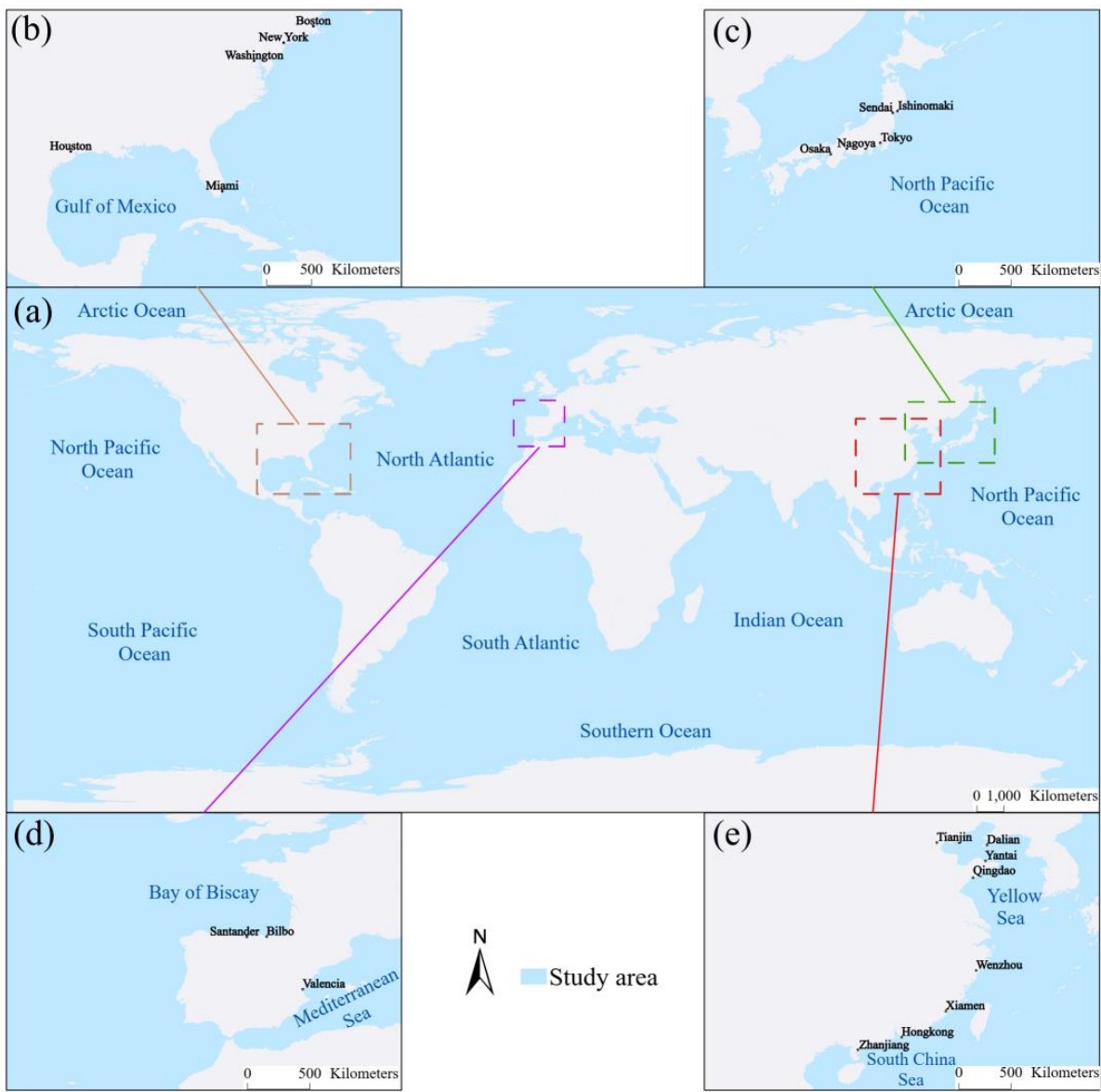

**Figure 1.** Study areas: (**a**) global ocean area; (**b**) United States of America offshore; (**c**) Japan offshore; (**d**) Spain offshore; (**e**) China offshore. Locations on the maps are referred to in the main text.

Global fishing effort can be mapped hourly based on a geographical grid at a spatial resolution of 0.1 decimal degrees [39]. However, we resampled the 0.01 degrees grid to $0.1 \times 0.1$ decimal degrees for the global scale analyses and used the 0.01 degrees grid for the state scale, after which monthly values were calculated by summing fishing hours in each grid cell. It should be noted that China, parts of the United States, and Spain adopted a lockdown policy from March to May 2020, but Japan did not implement lockdown measures during this period. For cultural and political factors, we selected the Chinese New Year and Christmas as cultural factors, and China's fishing closed season as the policy factor. To analyze the impact of culture and policy on fishing effort, we focused on the global fishing data from 2017 to 2019. Cultural and policy factors were mapped into AIS to calculate the proportion of fishing effort of Chinese fishing vessels and fishing vessels from other countries in each month.

To evaluate fishing gear, we used gear classes composed of: trawlers, trollers, fixed gear, set gillnets, squid jigger, pole-and-line, set longlines, dredge fishing, pots-and-traps, tuna purse seines, drifting longlines, other pure seines, seines, purse seines, other seines and "fishing" (remaining unknown classification of fishing gear after machine learning).

Set gillnets and drifting longlines were classified as longlines; integrated tuna purse seines, other pure seines, purse seines, other seines and seines were classified as purse seines; "Other" fishing gear comprised "fishing", trollers, fixed gear, squid jigger, pole-and-line, dredge fishing and pots-and-traps. We analyzed the distribution and trends for the following gear classes: trawlers, longlines, pure seines, gillnets and Other.

### 2.2. Monthly Chain Growth

Based on the concept of deviations from month-on-month growth, monthly chain growth [40] was used to determine monthly departures from an underlying linear yearly trend. Monthly changes of the target year were obtained from the monthly difference of the adjacent years before the target year. Firstly, we calculated the difference between the monthly data of each year and the previous year. Based on the results, we calculated the average of all differences except the difference between the target year and the previous year. The monthly chain growth was the difference between each month in the target year subtracted by the average difference between months in previous years. A positive number represents increasing departures from the trend, and a negative number represents decreasing departures, as calculated by Equation (1) below:

$$Monthly\ chain\ growth = M_{n-(n-1)} - M_{mean} \tag{1}$$

$$M_{mean} = \frac{\sum_{i=2}^{n-1} M_{i-(i-1)}}{n-1}$$

where $n$ represents the year of the data, $M_{n-(n-1)}$ represents the difference between a month in the target year and that month in the previous year, and $M_{mean}$ represents the average value of the difference between the monthly data in all years except the target year.

### 3. Results

### 3.1. Global Fishing Effort

AIS fishery effort for 2017 to 2020 (Figure 2) trended upward on average by 8.3% annually but declined in 2020 by 5.2% from 2019 to 2020. Monthly chain growth in 2020 (Figure 3) showed declines for every month with larger declines starting in January ($-0.7$ million hours) until a minimum in July ($-0.3$ million hours), but fishing activity dropped markedly in October ($-1.5$ million hours), which was the largest decline throughout the year, and then partially recovered in November ($-0.7$ million hours) and December (less than $-0.4$ million hours).

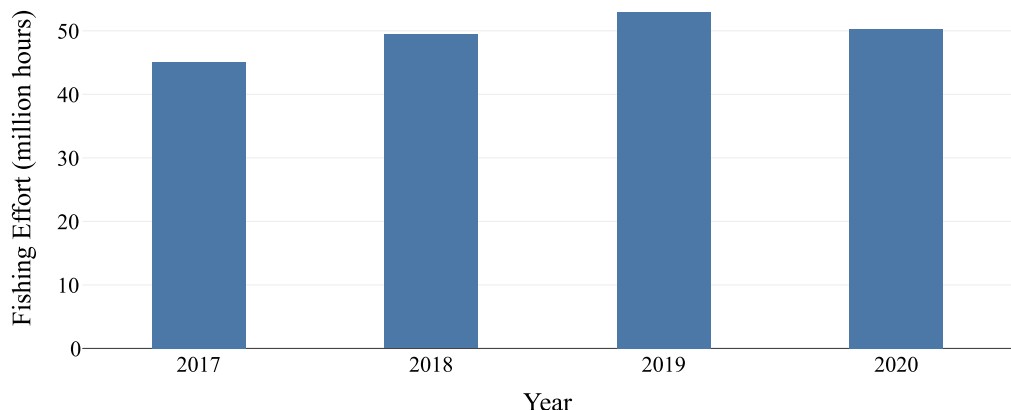

**Figure 2.** Global fishing effort from 2017 to 2020.

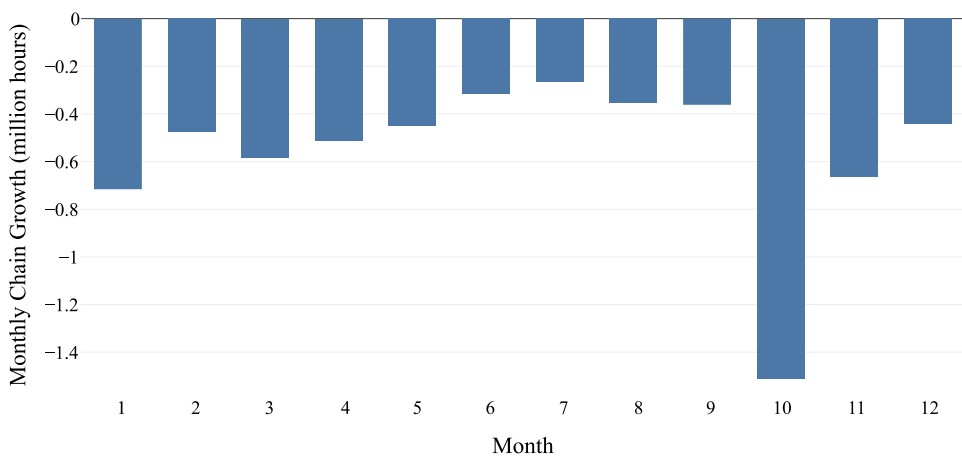

**Figure 3.** Monthly chain growth of global fishing effort between 2020 and previous year (2017 to 2019).

AIS fishing effort (Figure 4) was consistent with the findings of FAO [41], with most areas experiencing 0–50 h of effort and most effort in the high seas outside the Exclusive Economic Zone (EEZ) of coastal states. Data for small-scale fisheries in Southeast Asia were noticeably missing (due to a lack of AIS data for vessels under 15 m length) and this is a key limitation of our study. High fishing effort was mostly located along coastal areas of European countries, offshore areas of China and parts of the Pacific Ocean high seas. Coastal effort was banded or block-like, and annual effort in 0.1° grids was 100 h or more.

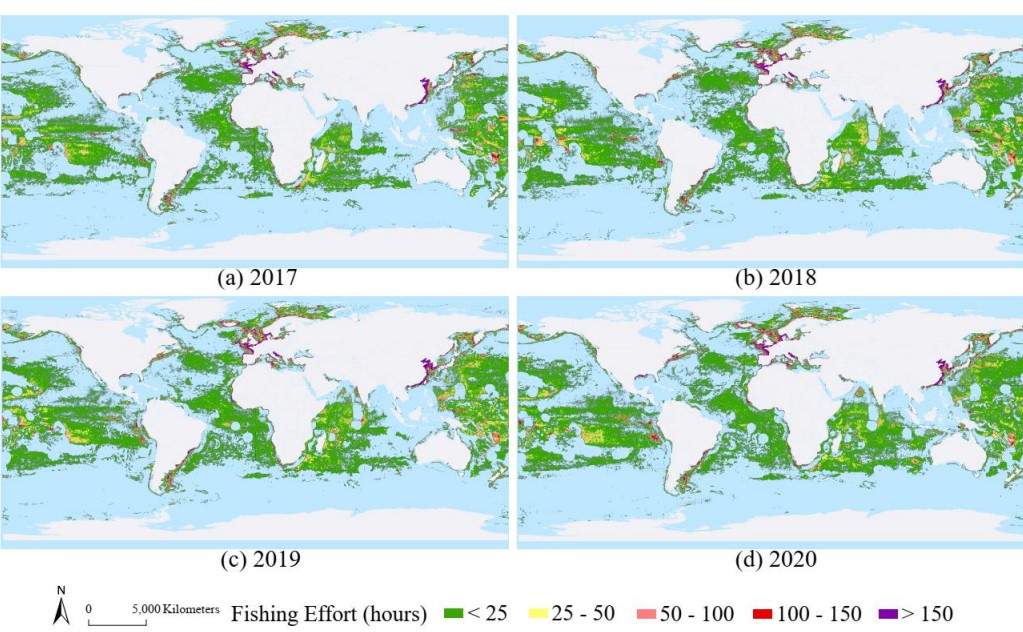

**Figure 4.** Spatial distribution of global AIS fishing effort from 2017 to 2020, color-coded according to the classes of fishing effort (hours).

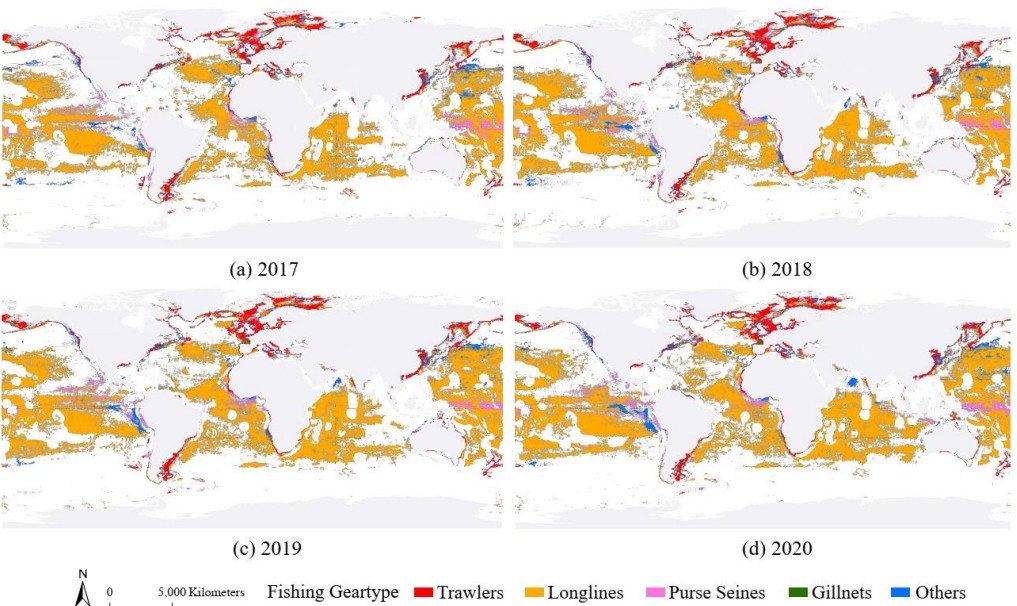

**Figure 5.** Spatial distribution of global fishing gear from 2017 to 2020.

Global fishing effort for 0.1° grids (Table 1 and bearing in mind the lack of small-scale fishing data) showed the percentage of grids fished increased yearly (2017–2019), and on average occurred within 25.67% of ocean grids (2017–2020), mostly from central Atlantic and eastern Pacific. However, the proportion of grids fished increased to 27.49% in 2020, despite the overall decline of effort noted in Figure 2, implying that fishing was more widespread but less intense.

**Table 1.** Percentage of gridded global ocean area fished in 2017–2020. Note that each grid is considered to contain effort regardless of the actual effort (hours) or the number/type of vessels or fishing gear within the grid (see discussion of Kroodsma et al. [22]).

| Year | 2017 | 2018 | 2019 | 2020 |
|------|------|------|------|------|
| 0.1° grid percentage | 23.91% | 26.29% | 26.81% | 27.49% |

The spatial distribution and trend of AIS fishing gear from 2017 to 2020 (Figures 5 and 6, Table 2) showed the world's primary fishing gears were trawlers and longlines each year. Most trawling efforts were in offshore EEZ areas distributed in strips along the coastline, while longlines were set mostly in the high seas (which refers to waters beyond EEZ, territorial seas and internal waters of non-coastal countries) and across large areas. Pure seines were mainly used in the mid-latitude offshore and open sea areas, gillnets dominated high-latitude offshore areas, and Other fishing gears were widely distributed across offshore EEZ areas.

**Table 2.** Cell grid counts of global fishing gear classes from 2017 to 2020.

|  | 2017 | 2018 | 2019 | 2020 |
|------|------|------|------|------|
| **Trawlers** | 121,697 | 128,107 | 128,940 | 124,525 |
| **Longlines** | 771,091 | 869,463 | 893,790 | 906,221 |
| **Pure seines** | 73,623 | 66,103 | 66,214 | 74,022 |
| **Gillnets** | 6216 | 6832 | 7261 | 7068 |
| **Other** | 75,831 | 76,477 | 72,121 | 120,823 |

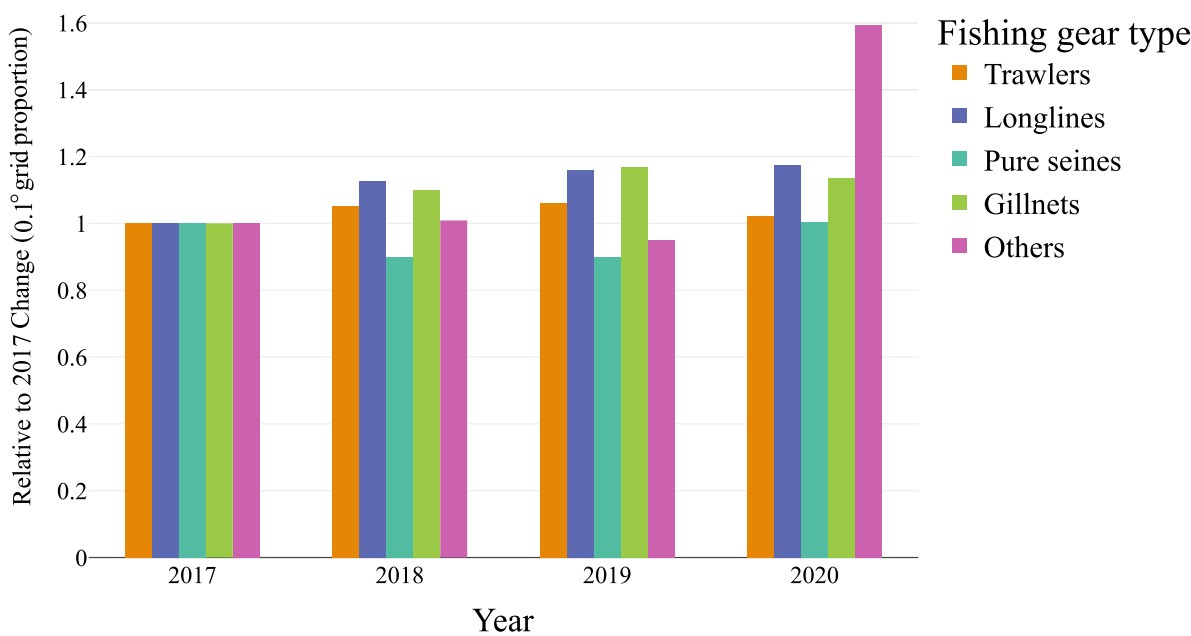

**Figure 6.** Yearly variation from 2017–2020 of the ratio of global grid counts by fishing gear type relative to 2017.

The proportion of trawlers and longlines in the annual fishery increased slightly each year (2017–2019), which is possibly related to overfishing policy restrictions on inshore fishery resources of some countries. However, in 2020, compared to the previous three years, the proportion of trawlers in 2020 declined the most, and Other fishing gears generally increased, specifically the Other category. The increase in Other gear was primarily associated with increases in squid jigger, pole-and-line and unknown fishing gear types (Figure 7).

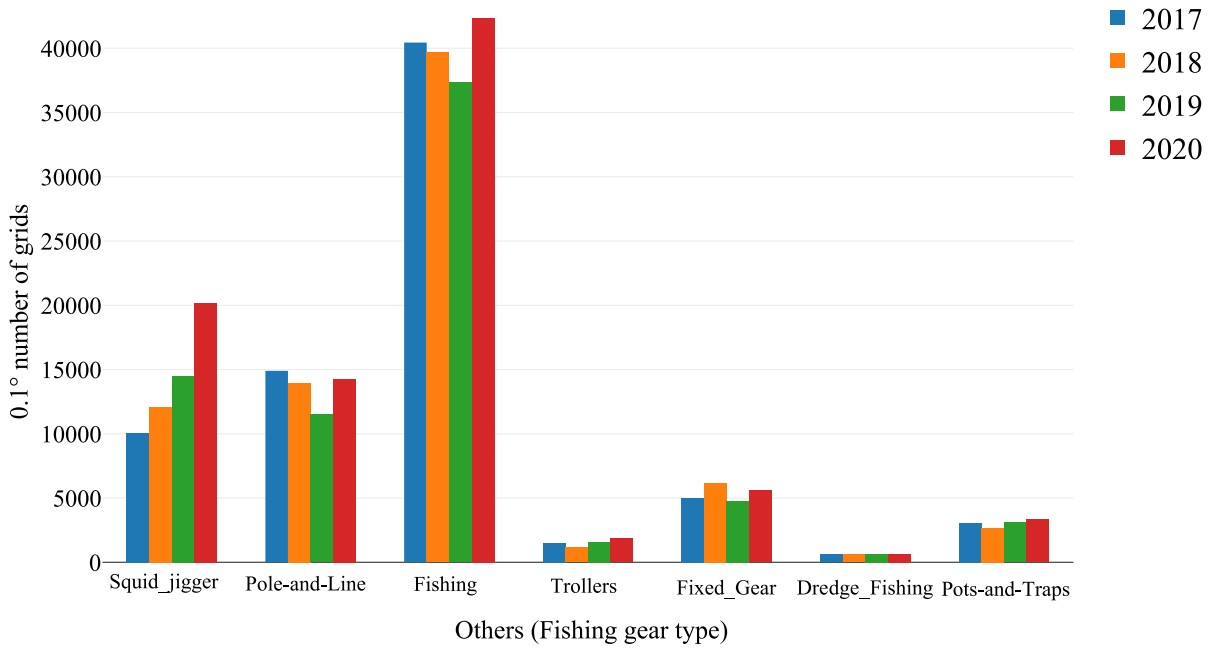

**Figure 7.** Yearly change in the global grid counts of fishing gear type in the "Other" class of gears.

### 3.2. Impact of Policy and Culture on Fishing Effort 2017–2019

Monthly variations in global fishing effort during 2017 through 2020 were stable (Figure 8), with the lowest and highest fishing effort in February and September, respectively. Culture

affected the fishing effort in February and December of each year, and policy affected effort from May to September. The start of cultural and policy factor events caused an average 33.45% decrease in effort per year compared to normal months followed by a recovery trend, especially during China's fishing closed season (Figure 8). The growth rate during China's fishing closed season in 2020 was 4.63% higher than that of the previous three years, but the maximum annual fishing effort decreased by 5.39% compared to 2019.

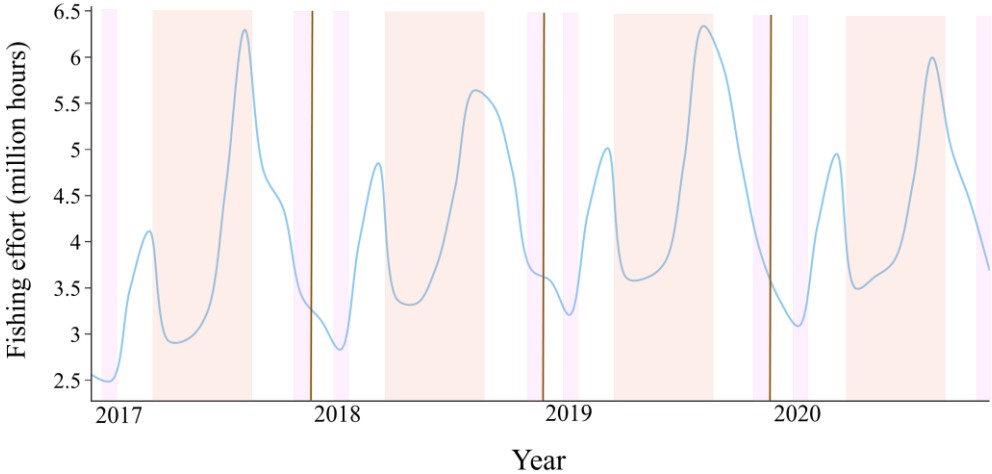

**Figure 8.** Global monthly fishing effort (millions of hours) from 2017 to 2020. The pink area is the cultural month of New Year and Christmas, and the orange area is China's fishing closed season.

We calculated the percentage of effort affected by these factors for China compared to other countries during relevant months (Feb and Dec for culture, and May to September for policy) from 2017 to 2019 (Figure 9). As shown in Figure 9a, fishing effort under culture for the other representative countries were basically stable in February every year, but fishing effort increased for China. However, the overall proportion of fishing operations was the lowest in February each year. The proportion of Chinese fishing vessels in December fluctuated and was similar to effort from other countries.

Figure 9b displays the proportion of fishing effort of Chinese fishing vessels and those of other countries in different months under the influence of policy factors. The main policy factor is China's fishing closed season from May to September every year. During this period, the annual average proportional fishing effort of Chinese fishing vessels was about 12.83%, and the total proportion decreased initially and then rebounded. In addition, the proportion of fishing effort of Chinese fishing vessels declined sharply from June to August every year.

China's fishing closed season was mainly concentrated in the EEZ, with the greatest impact in May and September. The average proportion of fishing effort of other countries during this period was about 25.44% and stable. As shown in Figure 9, although the proportion of fishing effort of Chinese fishing vessels decreased during China's fishing closed season in 2018, the overall fishing effort was still higher than 2017. Thus, the fishing effort of other countries during this period increased significantly compared to 2017.

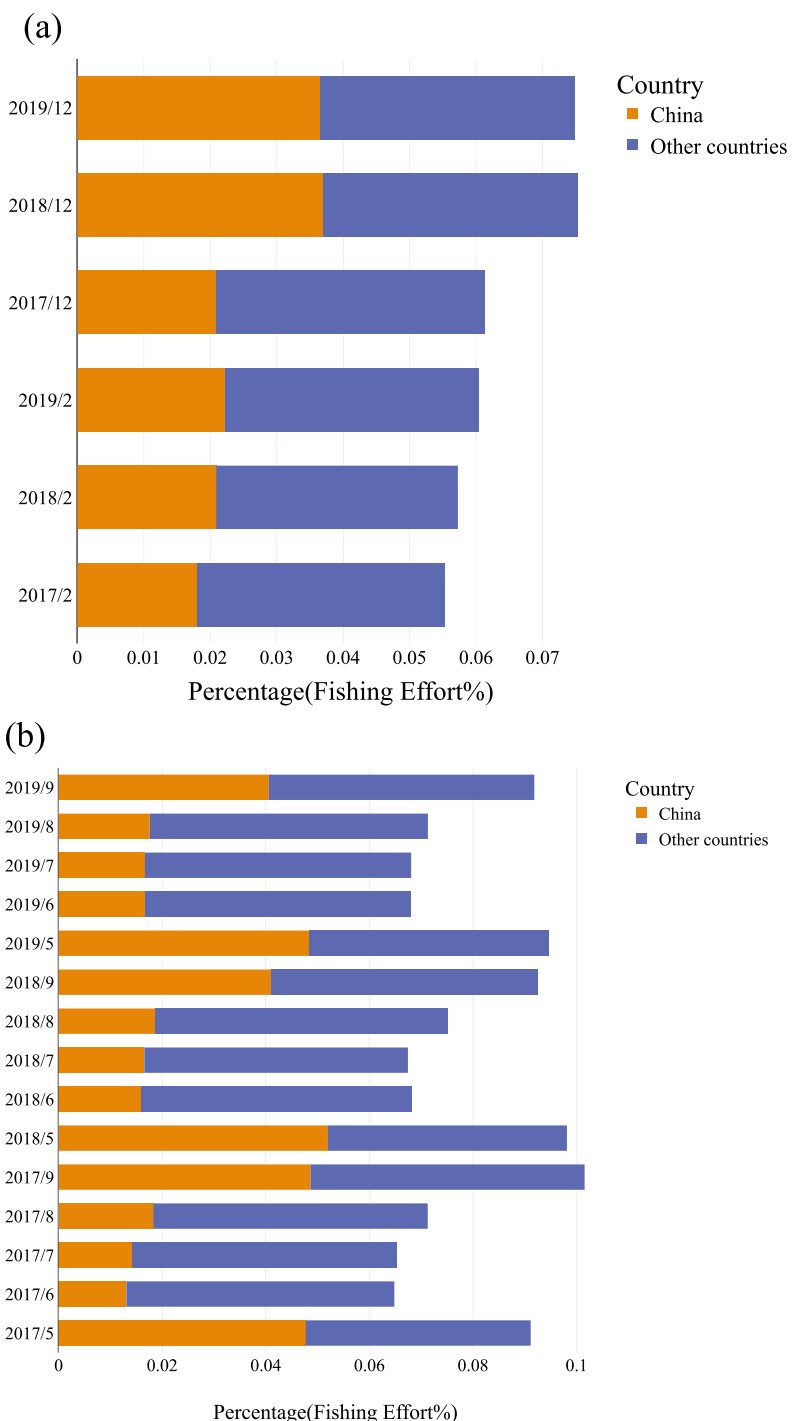

**Figure 9.** The impact of culture and policy on the percentage of fishing effort of China and other countries. (**a**) Impact of culture on fishing effort, (**b**) impact of policy on fishing effort.

Further analyses were conducted by analyzing the impact of Chinese New Year on China's fishing effort and Christmas' impact on the fishing effort of all countries except China. The influence of culture and policy on this attribution of fishing effort by culture and policy was 5.76% and 12.83%, respectively. Hence, overall policies had a greater impact on monthly fishing effort. The average monthly proportion of fishing effort in the two months affected by culture and the five months affected by policy was similar at 2.88%, and 2.84%, respectively.

Table 3 and Figure 10 show the proportion of fishing gear in China and other countries associated with culture and policy during 2017 through 2020. During the February Chinese New Year from 2017 to 2019, proportions of trawlers, pure seines, gillnets and Other fishing gear declined each year; only longlines increased to over 50%. The Chinese New Year therefore had a greater impact on fishing vessels in the EEZ area, and fishing operations were mostly in the high seas. In 2020, trawlers and longlines fluctuated significantly compared to the previous three years; possibly due to the implementation of China's epidemic policy, trawlers in the EEZ decreased significantly. At the same time, high seas vessels could not enter port, and longlines showed an upward trend. During the Christmas season in December, trawlers dominated over longlines—the reverse of the typical pattern in February. Annual fluctuations of Other fishing gears were relatively small, but the main inshore fishing by trawlers declined during 2017 through 2019, and gillnets trended upward. In 2020, the trend reversed owing to the impact of the epidemic.

**Table 3.** Proportion of fishing gear under the influence of culture and policy on fishing effort in China and other countries.

| | Month | Trawlers | Longlines | Purse Seines | Gillnets | Other |
|---|---|---|---|---|---|---|
| **2017** | China (February) | 27.01% | 45.44% | 1.86% | 6.85% | 18.84% |
| | China (May–September) | 9.78% | 70.69% | 2.13% | 3.34% | 14.07% |
| | Other countries (December) | 53.48% | 31.29% | 2.47% | 3.55% | 9.21% |
| **2018** | China (February) | 25.44% | 52.38% | 0.99% | 5.67% | 15.51% |
| | China (May–September) | 10.89% | 71.01% | 1.77% | 4.47% | 11.87% |
| | Other countries (December) | 50.27% | 32.54% | 2.89% | 3.74% | 10.55% |
| **2019** | China (February) | 23.45% | 54.07% | 0.99% | 5.59% | 15.9% |
| | China (May–September) | 11.75% | 67.26% | 1.54% | 5.03% | 14.41% |
| | Other countries (December) | 51.90% | 31.9% | 2.60% | 3.84% | 9.76% |
| **2020** | China (February) | 19.95% | 60.53% | 1.02% | 3.75% | 14.75% |
| | China (May–September) | 29.21% | 41.07% | 1.36% | 6.62% | 21.72% |
| | Other countries (December) | 53.09% | 29.32% | 2.81% | 3.5% | 11.29% |

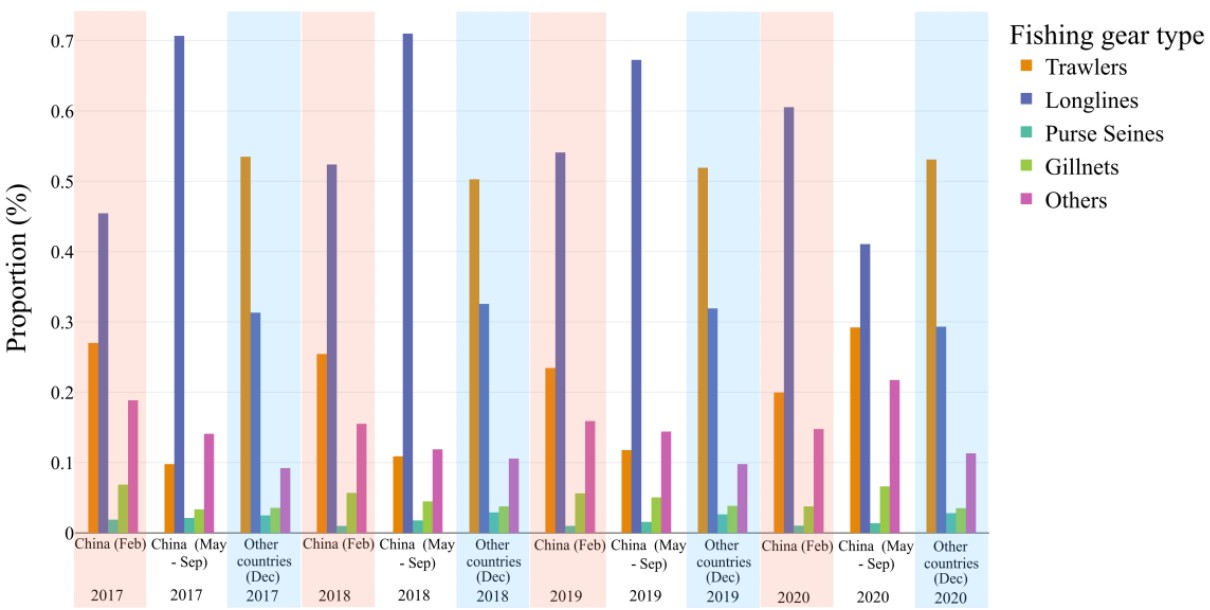

**Figure 10.** Comparative plots for 2017 to 2020 of the proportion of fishing gear under the influence of culture and policy on fishing effort in China and other countries for specific months (Feb and Dec for culture; May to Sep for policy).

In 2020, given the ongoing epidemic, high seas fishing from all countries declined, and fishing operations were mainly concentrated in the EEZ of each country, leading to a rebound in trawling effort. Thus, Christmas had little impact on fishing methods in other countries outside of China, and the main affected area was offshore of the EEZ waters. During China's fishing closed season, Chinese fishing vessels accounted for only a small proportion of all fishing gear, except for longlines. During this period, trawlers and gillnets increased by year, while longlines and pure seines declined in the high seas. The 2020 year was particularly prominent; in view of the impact of policies and the global epidemic, high seas fishery operations declined by around 30% compared with the previous three years, and about 20% was due to trawlers.

### 3.3. The Influence of COVID-19 Restrictions on Fishing Effort

In 2020, lockdown in several countries caused by the novel coronavirus (COVID-19) pandemic impacted global fishing effort. As shown in Figure 11, the fishing effort curve in 2020 was similar to previous years (2017–2019; (Figure 8)), but fishing effort decreased sharply every month. Prior to the lockdown in various countries, the fishing effort in January was lower than previous years. The world's first lockdown was imposed in Wuhan, China, on 23 January 2020, while China officially entered its national lockdown phase after the Chinese New Year. The global fishing effort in February and March increased slowly due to the impact of the lockdown. The World Health Organization (WHO) declared the COVID-19 lockdown a global emergency on March 11, and most countries announced their policies and entered their March–May first lockdown phase. The fishing effort trend was similar to previous years, but the decline was rapid, and extreme values were lower than those in 2019. Although the second lockdown was implemented in some countries after June 2020 and had an impact on the fishing effort, the timing was not uniform, and the temporal scale was less than the monthly scale we studied.

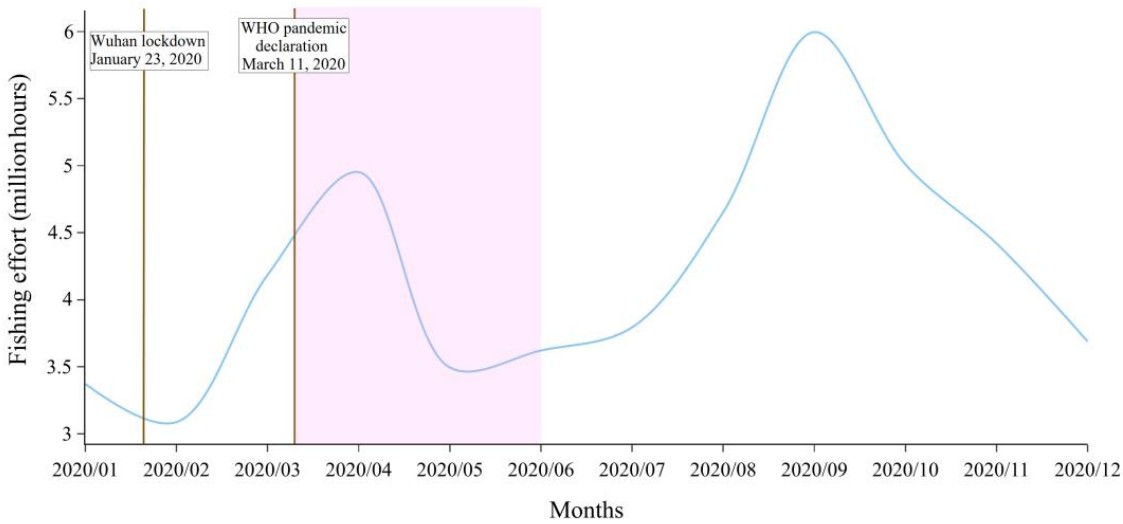

**Figure 11.** Fishing effort in 2020 (pink area represents the time for the first lockdown in most countries around the world).

To further analyze the impact of lockdown policies on fishing effort and gear, we calculated the difference in the spatial distribution of offshore fishing effort and fishing gears in the four countries in 2020 (Figures 12 and 13, respectively). The spatial distribution of fishing effort and fishing gears (Figures 12a and 13a, respectively) during the lockdown period from February to March in China was dominated by trawlers. Fishing effort in most coastal areas was limited, and there was no fishing activity in the coastal areas of Hainan Island. The fishing effort in major ports to the west of the coastal areas was more than 150 hours, of which the main fishing gear consisted of trawlers.

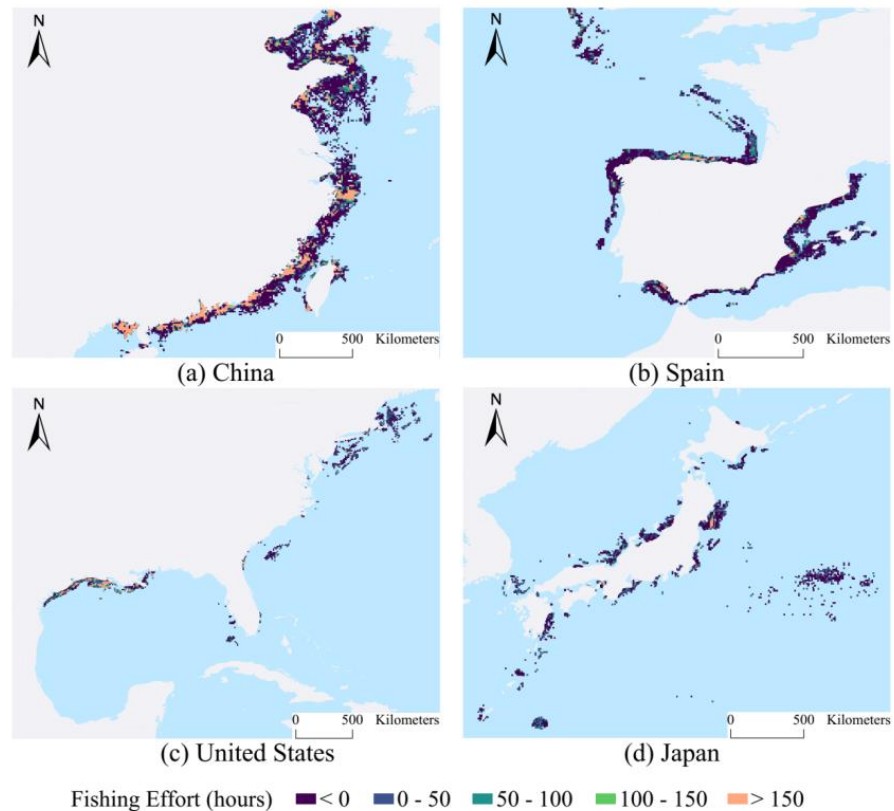

**Figure 12.** Lockdown and non-lockdown countries' fishing effort (hours) spatial distribution from March to May 2020.

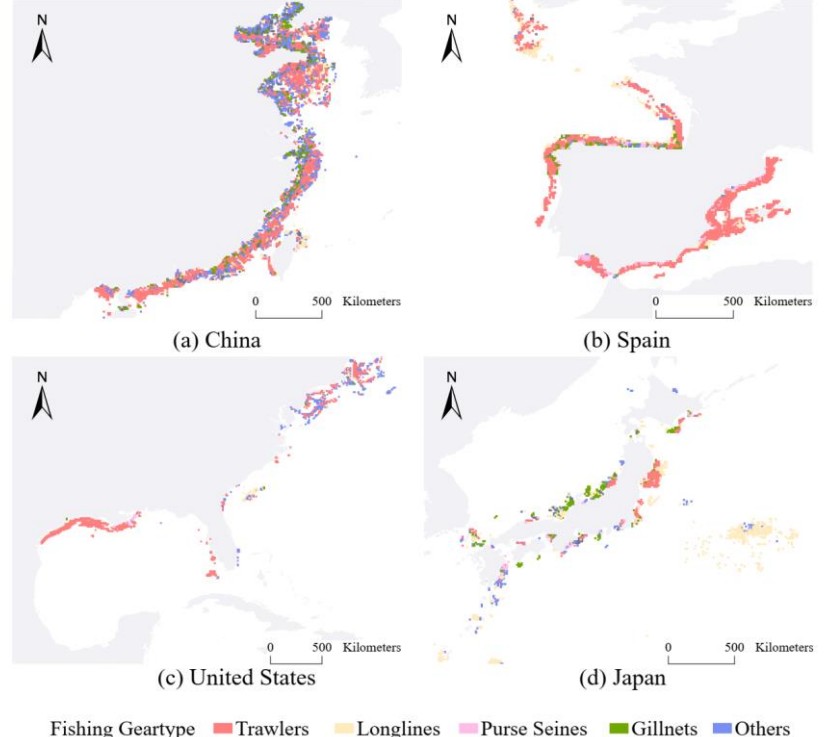

**Figure 13.** Lockdown and non-lockdown countries' fishing gear spatial distribution from March to May 2020.

Figures 12b and 13b show spatial variations in fishing effort for Spain during its first lockdown, when the main fishing gear was trawlers. The fishing effort was absent along most of the coastal areas in the northwest and southeast of Spain, but there were some areas of concentrated offshore gillnet operations. For instance, there were around 100 fishing hours in the Santander offshore area which showed a block distribution.

Figures 12c and 13c show that the main fishing areas of the eastern United States' lockdown were the Gulf of Mexico and offshore of New York and Boston, where the fishing gear was mainly trawlers and Other fishery operations. Fishing effort was limited due to the lockdown in New York, and most offshore areas displayed limited fishing effort. In the Gulf of Mexico around Houston where lockdown was not implemented, the range of offshore fishing activities was reduced compared to previous years. In contrast, fishing effort was more than 100 h around New York and Boston.

Figures 12d and 13d show spatial variations across Japan's offshore areas, which had not implemented a lockdown during this period. Fishing activity area reduced sharply for the mix of fishing gear in the coastal waters of Japan. The northeastern reduction in fishing effort was mainly due to trawlers, and the northwest reduction was from offshore gillnets. Most fishing effort in the area was limited, but fishing effort in the Ishinozaki and Osaka offshore areas was more than 100 h.

Monthly chain growth for China, Spain, the United States and Japan (Figure 14) all showed negative growth in January, and except for the United States, fishing effort in every month decreased compared to the previous three years. In Figure 14a, China's fishing effort during the February and March national lockdown period decreased dramatically compared to the previous three years and increased only in April. In addition, during China's fishing closed season from May to September, the decline was minimal. Although the fishing closed season was finished in October, fishing volume decreased significantly compared to the previous three years and rebounded from November to December.

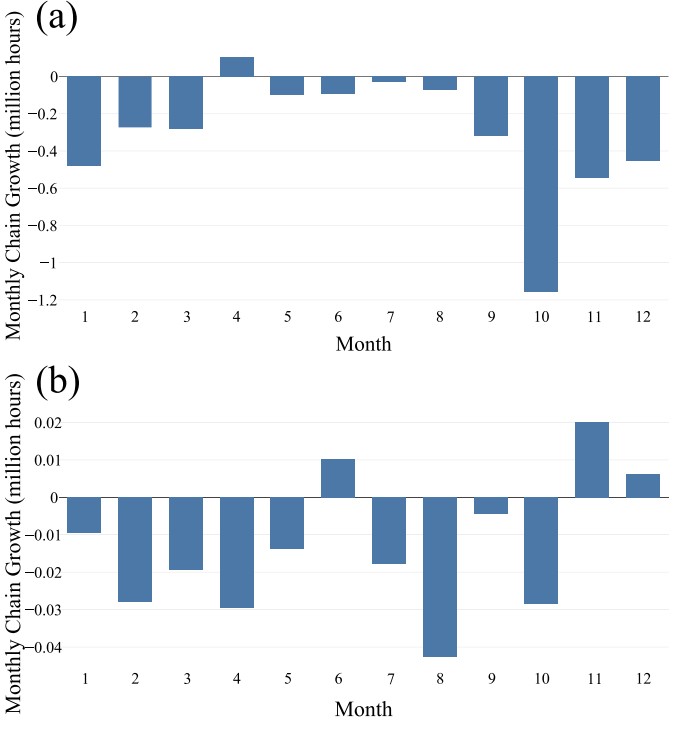

**Figure 14.** *Cont.*

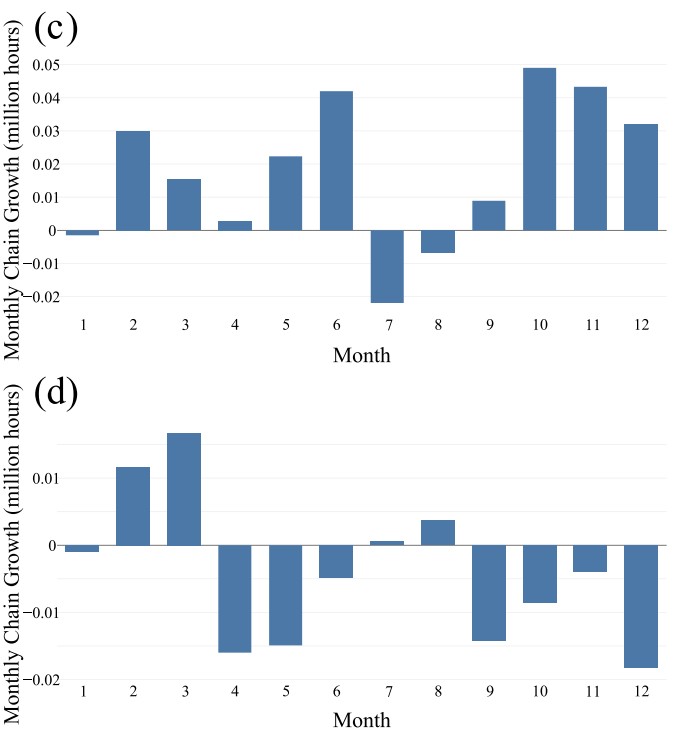

**Figure 14.** Monthly chain growth from 2017 to 2020 in some countries: (**a**) China; (**b**) Spain; (**c**) United States; and (**d**) Japan.

In Figure 14b, Spain's fishing effort declined for the first five months before recovering to positive growth after the end of the first lockdown in June, and then effort declined again from July to October, with a minimum in August. In November and December, it gradually returned to positive growth.

In Figure 14c, except for January, July and August, which showed negative growth, the fishing effort in other months of the United States in 2020 showed a positive growth trend compared with the previous three years, and the positive growth in October reached the highest level of the whole year.

In Figure 14d, Japan's fishing effort grew from January to March but dropped sharply from April to June. It rebounded slightly from July to August, but declined sharply every month from September to December; reaching its lowest point in December.

## 4. Discussion

We analyzed and compared cultural and policy drivers of fishing effort and fishing gear monthly during 2017 through 2019 and then overlaid impacts from COVID-19 lockdowns for 2020. We found the overall monthly cyclical pattern was similar across all years, and there was a yearly increase from 2017–2019, but a significant sharp decline in 2020 [42–44]. Each year, cultural and policy factors reduced fishing effort, which was also reflected in the proportion due to these factors, but the decline associated with policy factors exceeded about 50%. Policy and culture only affect the proportion of fishing effort in some specific months during the year and have little impact on fishing activity over the whole year. The annual increase in fishing effort from 2017 to 2019 showed most countries did not effectively implement policies for the protection and sustainable development of marine fishery resources. China's fishing ban policy reduced longline operations of Chinese fishing vessels, which led to a decline in fishing effort in the high seas.

In 2020, due to the impact of COVID-19, the overall fishing effort declined throughout the year, with the most severe decline to trawlers [45], followed by gillnets and longlines. Purse seines, however, increased, possibly because EEZ areas were restricted by lockdown and fishery activities were forced to the high seas, which also experienced an increase in Other fishing methods and fishing areas fished. In the four selected countries for the

COVID-19 study, fishing effort declined during the first lockdown period but rebounded rapidly during the remaining months. In summary, fishing effort in 2020 was affected by COVID-19 to varying degrees for all months and countries, resulting in a general overall global decrease of fishing effort in 2020. Compared with culture and policy, global emergencies had a more significant impact on fishing effort.

Overfishing directly affects fisheries and can cause other flow-on impacts [46], such as environmental pollution, bycatch of threatened species and ecological impacts, which may undermine sustainable development of regional fisheries and marine ecosystems health. Global fishing effort is not only concentrated in the EEZs of various countries, but also in the North Pacific Ocean and South Pacific Ocean, indicating high-seas fishing is increasing for various nations.

Our results are exploratory, but future explanatory casual analyses may be possible using data for other factors, such as trade, labor and related economic policy changes from COVID-19 lockdowns. Moreover, spatio-temporal analyses would benefit from improved and more comprehensive data monitoring. Currently, fishing vessels are discriminated from radar images and identified using Random Forest classification [47]. Improvements are possible by combining AIS and remote sensing data to increase the effectiveness of fishing vessel activity monitoring. The enhanced monitoring will help in the implementation and monitoring of fishery policies [48–51]. Although AIS and remote sensing data can complement each other, further improvements may be possible through sub-regional scale studies, but it is difficult to find other data sources that can effectively complement AIS on a global scale.

The four countries we selected are well represented in AIS data [52,53] and also account for a high proportion of FAO fish catches. It should be noted other activities can impact fishing effort [54], such as from recreational and coastal fish farming activities. Overall, our results demonstrate that culture, policy and global emergency had a significant impact on spatio-temporal distributions of global fishing effort.

Currently, AIS fishing vessel activities are divided into industrial and small-scale fisheries for vessels with ship length less than 15 meters [55]. However, existing AIS data are mainly for industrial fisheries; hence, operations in Southeast Asia comprising mainly small-scale fishing are unavailable (Figure 4). Attempts have been made to collate and harmonize small-scale fishery data from a range of public sources from 1950 to 2014 mapped to 30 min spatial cells [56]. However, the time scale is not appropriate, and most of the data are in the form of yields from other marine species, such as shrimp and all kinds of economic fish. To calculate fishing effort, it is necessary to redefine the calculation formula for fishing effort. We used fishing time as fishing effort, but the two data sources are different. Data magnitude errors are therefore likely to occur when they are combined. In future, small-scale fisheries could be monitored from remote sensing images and other data [57,58], which will facilitate analyses of cultural, political, and global emergency drivers of small fisheries in Southeast Asia.

Overall, despite some limitations, this is the first quantitative global assessment of the interacting effects of culture, policy and global emergency of COVID-19 on fishing effort and gear. In addition, we demonstrated open-source AIS data has the advantage of replicability and knowledge exchange of fishing activities.

## 5. Conclusions

We showed that data processed by the GFW enable the study of culture, policy and global emergencies drivers on the global fishing effort. Annual global fishing effort increased from 2017 to 2019, concentrated in the territorial waters of European countries, part of the high seas in the Pacific Ocean and the territorial waters of China, which were dominated by trawler and longline operations. Fishing in the shallower seas of Southeast Asia were noticeably lacking due to the absence of AIS data for vessels less than 15 m in length; this needs to be addressed in the future to facilitate more comprehensive analyses. In the first three years, without global emergencies, the annual average monthly fishing

effort accounted for about 2.88% for months influenced by culture and about 2.84% for months subjected to policies. Policies had greater impact on the global fishing effort than culture, and Chinese longlines decreased each year. Analyses of four countries with lockdown and without lockdown policies showed declines greater than culture and policy, but fishing in the US saw positive growth for most months. Compared with the previous three years, the decline in fishing gear in the four countries was mainly due to trawlers. Our analyses and results provide an alternative and more accurate perspective of drivers and controls of fishing effort, which can guide global marine spatial planning and sustainable fishery management.

**Author Contributions:** Conceptualization, B.H. and F.Y.; methodology, B.H., F.Y. and F.S.; software, B.H. and H.Y.; validation, V.L., B.H. and Y.C.; formal analysis, L.K.; data curation, L.K. and B.H.; writing—original draft preparation, B.H.; writing—review and editing, B.H., H.Y., F.S. and V.L.; visualization, B.H. and V.L.; supervision, W.W.; project administration, F.Y. All authors have read and agreed to the published version of the manuscript.

**Funding:** This research was supported by the National Natural Science Foundation of China (41890854), the President's International Fellowship Initiative of Chinese Academy of Sciences (2020VEA0009).

**Data Availability Statement:** Not applicable.

**Acknowledgments:** I am very grateful to the reviewers and editors for their suggestions on the revision of the manuscript, and to other colleagues in the laboratory for the exchange of the content of the manuscript, and special thanks to Vincent Lyne for his assistance in the revision of the manuscript and the English writing.

**Conflicts of Interest:** The authors declare no conflict of interest.

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
