# Peer review of "Global Fisheries Responses to Culture, Policy and COVID-19 from 2017 to 2020"

_remotesensing, doi:10.3390/rs13224507_

Round 1

Reviewer 1 Report

See document

Author Response

Dear reviewer 1, I had put the new manuscript and your response in the compressed package, and the new manuscript has marked the location of the response to your comments. Thank you for reviewing and suggesting my manuscript. I wish you all the best!

Reviewer 2 Report

This is an interesting paper. My main question is: how can we ensure that there were not other aspects influencing fishing in 2020? Was it the sole results of lockdowns? or maybe of labour availability? or maybe of death/sickness amongst employees? or maybe of economic policies to unemployed? or maybe a result of decreased food trade? see the paper of Mohammad al Saidi on the nuances of similar analysis: Al-Saidi, M., et al. (2021). The water-energy-food nexus and COVID-19: Towards a systematization of impacts and responses. Science of The Total Environment779, 146529.

Author Response

Dear reviewer 2, I had annotated in the new manuscript. Thank you for reviewing and suggesting my manuscript. I wish you all the best!

Round 2

Reviewer 1 Report

Attached

Author Response

Dear reviewer 1, thank you for commenting on my manuscript again. We have already responded to your comments in the Word version. I wish you all the best!

Reviewer 2 Report

fine

Author Response

Dear reviewer 2, thank you for your affirmation of the comments on my manuscript. I wish you all the best!